# An Indoor and Outdoor Multi-Source Elastic Fusion Navigation and Positioning Algorithm Based on Particle Filters

Guangwei Fan [1,2], Chuanzhen Sheng [1,2,*], Baoguo Yu [1,2], Lu Huang [1,2] and Qiang Rong [1,2]

1   The 54th Research Institute of China Electronics Technology Group Corporation, Shijiazhuang 050081, China; fgweihb@163.com (G.F.); yubg@sina.cn (B.Y.); 18642720668@163.com (L.H.); rong_54@163.com (Q.R.)
2   State Key Laboratory of Satellite Navigation System and Equipment Technology, Shijiazhuang 050081, China
*   Correspondence: shengchuanzhen@163.com; Tel.: +86-136-3311-5269

**Abstract:** In terms of indoor and outdoor positioning, in recent years, researchers at home and abroad have proposed some multisource integrated navigation and positioning methods, but these methods are navigation and positioning methods for a single scene. When it comes to the switching of indoor and outdoor complex scenes, these methods will cause the results of position with a marked jump and affect the accuracy of navigation and positioning. Aiming at the navigation and positioning problem in the case of indoor and outdoor complex scene switching, this paper proposes a multisource elastic navigation and positioning method based on particle filters, which makes full use of the redundant information of multisource sensors, constructs an elastic multisource fusion navigation and positioning model after eliminating abnormal data, elastically gives different particle weights to multisource sensors according to environmental changes and realizes the elastic fusion and positioning of multisource sensors through filtering. The test results show that this method has high navigation and positioning accuracy, the dynamic positioning accuracy is better than 0.7 m and there will be no jumping of positioning results in the process of scene switching, which improves the navigation and positioning accuracy and stability in complex and changeable indoor and outdoor environments.

**Keywords:** indoor and outdoor positioning; elastic PNT; multisource fusion; particle filter

## 1. Introduction

Although the PNT service provided by GNSS has the characteristics of all-weather, all-time and global coverage [1], GNSS also has obvious weaknesses, such as GNSS signal being easy to be obscured, interfered and deceived, resulting in the security and integrity of PNT service [2,3]. Any single PNT information source may have risks. PNT services involving personal safety must be safe and reliable. Therefore, the utilization of "redundant" PNT information is very important in complex environments [4].

The fundamental significance of the research on elastic PNT navigation and positioning technology is to help the satellite navigation system, make full use of the redundant information of multisource sensors, improve the reliability, safety and robustness of indoor and outdoor dynamic navigation and positioning and form new services that are not available in the current satellite navigation system. Therefore, the significance of "elastic PNT navigation and positioning technology" is to solve the problem that a single satellite navigation terminal cannot realize continuous, stable and reliable submeter real-time location service indoors and outdoors in complex environments [5]. The precise and seamless space-time location service provided by multisource elastic integration positioning will become an important benchmark to measure the smart economy and smart society. The multisource elastic fusion positioning technology will promote PNT from single, individual and regional to comprehensive, group and global. In the future, the multisource elastic fusion positioning network will develop towards IOT and intelligence. Multi-source elastic

fusion positioning technology has attracted extensive attention in industry because of its high positioning accuracy and high robustness. The problem of multisource elastic fusion location technology mainly focuses on the design of the fusion algorithm. The following problems must be considered in the positioning design of multisource elastic fusion [6]: a dynamic target is tracked by multiple different sensors and each sensor has different measurement results and noise characteristics. We must also consider how to dynamically adjust the weights of measurement results of multiple sensors according to environmental changes to make the final fusion estimation result better than that of a single sensor or more stable positioning [7].

Domestic and foreign experts have done a lot of research and published many papers on indoor and outdoor multisource fusion seamless positioning. For example, at the beginning, people used inertial sensors to obtain pedestrian heading and step information outdoors and fused with satellite navigation to improve the stability of navigation and positioning [8,9]. However, when satellite navigation is not available, the integrated positioning of inertial navigation and satellite navigation cannot maintain high navigation and positioning accuracy for a long time. The combination of communication base stations and satellite navigation is used to solve the navigation and positioning problem in urban canyons [10–12], which can solve the outdoor blind supplement navigation and positioning problem of satellite navigation, but it cannot solve the indoor navigation and positioning problem. In indoor navigation and positioning, Bluetooth is used for navigation and positioning, and its coverage and positioning accuracy are related to the deployment number of Bluetooth modules. Generally, the deployment interval of Bluetooth modules with positioning accuracy of 1 m is about 1 m [13–15], and the ability of large-scale promotion is poor. Most of the existing mobile communication devices, including wireless local area networks (WLANs), smart phones and laptops, are embedded with Wi-Fi modules, which can be used to build indoor Wi-Fi positioning systems [16,17]. However, Wi-Fi positioning usually requires the positioning method of fingerprint comparison, and the fingerprint database needs to be established in advance, the workload is large, and the positioning accuracy is about 3–5 M. Optical navigation is also a new navigation method rising in recent years, but optical navigation can only provide relative position and is greatly affected by light [18,19]. Pseudolites can be used in indoor and outdoor navigation and positioning and have good navigation and positioning accuracy, but the positioning performance in indoor and outdoor switching zones and indoor narrow and long zones is poor [20–22]. A single navigation source cannot solve the problem of navigation and positioning in indoor and outdoor complex and changeable environments. People have successively studied the navigation and positioning technology of multisource fusion [23–25]. However, in complex and changeable environments, the indiscriminate fusion of multisource sensor information will reduce the navigation and positioning accuracy of the carrier and affect the stability of navigation and positioning. Therefore, it is necessary to study the multisource elastic fusion navigation and positioning method.

At present, the research mainly focuses on the integration of satellite navigation, Wi-Fi, Bluetooth, 5G, inertial navigation, visual sensors and other navigation sources [7]. The amount of research on multisource elastic fusion navigation and positioning is relatively little, and how to make reliable use of elastic navigation resources is still unknown. This paper presents an indoor and outdoor multisource elastic fusion positioning algorithm based on particle filters. The algorithm flexibly selects the currently available sensor data and weights according to environmental elasticity to realize continuous and stable navigation and positioning in indoor complex environments. The innovative contribution of this paper is mainly reflected in the following aspects:

1. The relationship between inertial sensor data and Wi-Fi, Bluetooth and pseudolites is deduced, and the multisource sensor integrated positioning model is constructed;

2. An elastic navigation and positioning method based on particle filters is proposed, which can adaptively select the current multisource sensor data and sensor particle weight according to the environmental change, form the optimal elastic fusion positioning model

and solve the problem of seamless navigation and positioning in indoor and outdoor complex environments;

3. The indoor and outdoor complex environments are built, and a large number of experiments are carried out to verify the navigation and positioning performance of the proposed algorithm in complex and changeable environments.

## 2. Introduction of Elastic Navigation and Positioning System

### 2.1. System Composition

Using all available navigation resources, the elastic navigation and positioning terminal elastically integrates sensors such as the navigation receiver, inertial sensor, Wi-Fi, UWB, vision and lidar. The types of sensors can be flexibly selected according to the actual application scenarios. The optimal fusion location model is matched according to the environment and various prior information, and the sensor data are elastically selected for multisource fusion location to realize the accurate estimation of the carrier position, see Figure 1.

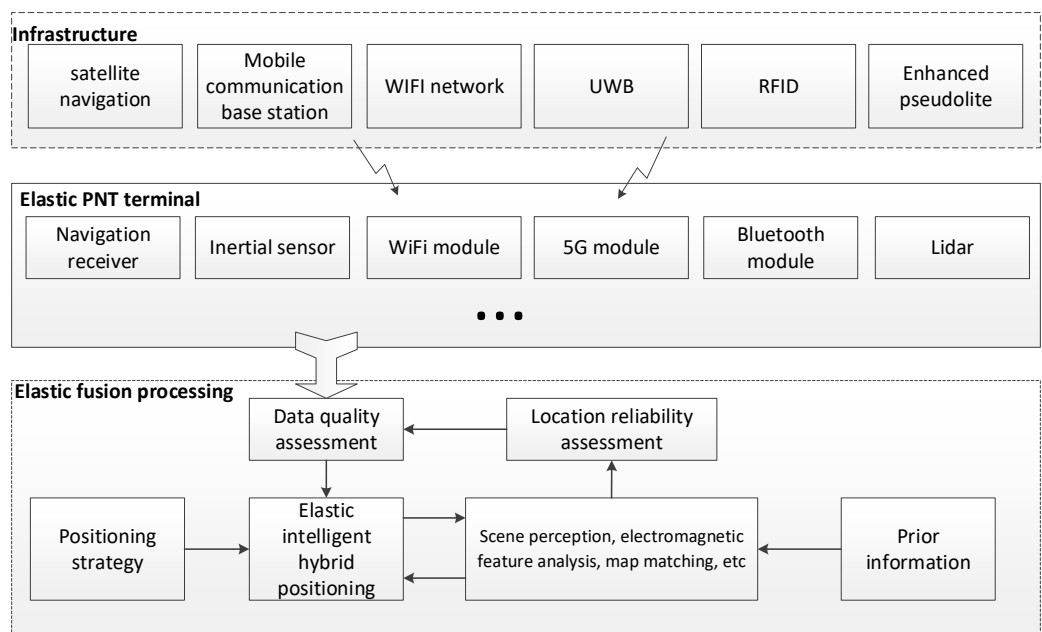

**Figure 1.** Elastic PNT positioning principle.

The elastic PNT terminal can receive the information of various navigation resources. This information is received and processed by the sensors in the elastic PNT terminal, and all data are unified under the same spatiotemporal benchmark. Then, the data quality of these sensor data is evaluated, and the optimal data combination involved in positioning is found in the redundant information for elastic intelligent hybrid positioning, The processing process of elastic intelligent positioning can be controlled by a cooperative positioning strategy and an a priori information model.

### 2.2. Receiving Terminal

Taking a centimeter-level positioning chip and module as the core, combined with inertial navigation, geomagnetism, Wi-Fi, Bluetooth and other sensors and integrated, an elastic PNT positioning terminal is developed. An elastic PNT positioning terminal is composed of a core module and an expandable module. The core module motherboard is mainly composed of an interface unit, an overvoltage and undervoltage protection unit, a lithium-battery-charging management unit, a voltage rise and fall conversion unit, a linear power conversion unit, a navigation and positioning module, an inertial device, a Bluetooth module, and Wi-Fi and 5G modules, and various expansion module interfaces

are reserved. The extensible modules mainly include: a UWB module, camera, lidar, etc. The composition principal block diagram of the terminal is shown in Figure 2.

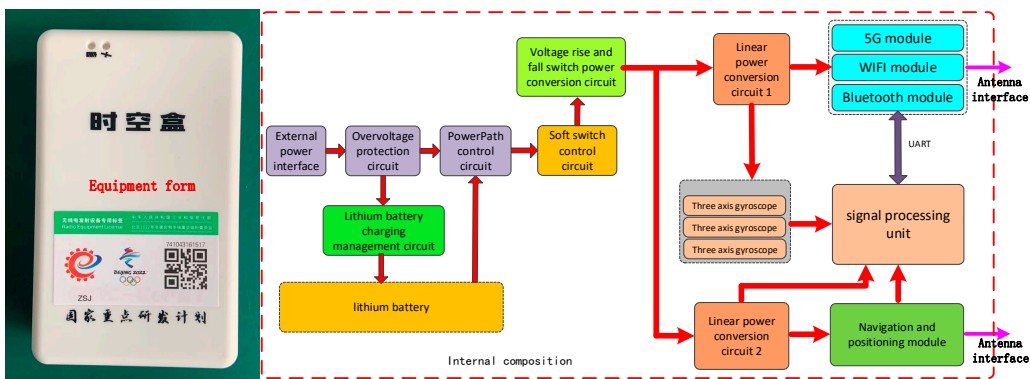

**Figure 2.** Composition block diagram of elastic PNT positioning terminal.

## 3. Sensor Positioning Principle

This section briefly introduces the navigation and positioning principles of several common sensors and gives the implementation steps.

### 3.1. Principle of Indoor Pseudosatellite Positioning

A pseudolite is usually fixed at a known position, and the position of the receiver can be obtained by using the pseudorange observation measurement. In theory, the position of the receiver can be obtained by using the simultaneous equations of the coordinates of three pseudolites and the receiver. However, in the process of practical application, the time between the pseudolite system and the receiver cannot be fully synchronized, and there is usually a certain clock deviation. Therefore, in order to calculate the coordinate position of the receiver, the clock difference needs to be taken into account as an unknown quantity. At least four pseudolites are needed to achieve positioning.

We assume that the pseudocode ranging and positioning equations of a pseudosatellite at time $k$ are as follows:

$$\begin{cases} r_{1k} = ct_{1k} + \delta t_{1k} + n_{1k} = \sqrt{(x_k - x_{s1}) + (y_k - y_{s1}) + (z_k - z_{s1})} + \delta t_{1k} + n_{1k} \\ \qquad\qquad\qquad \vdots \\ r_{ik} = ct_{ik} + \delta t_{ik} + n_{ik} = \sqrt{(x_k - x_{si}) + (y_k - y_{si}) + (z_k - z_{si})} + \delta t_{ik} + n_{ik} \\ \qquad\qquad\qquad \vdots \\ r_{Nk} = ct_{Nk} + \delta t_{ik} + n_{Nk} = \sqrt{(x_k - x_{sN}) + (y_k - y_{sN}) + (z_k - z_{sN})} + \delta t_{ik} + n_{Nk} \end{cases} \tag{1}$$

where $(x_k, y_k, z_k)$ is the carrier position coordinate at time $k$, $(x_{si}, y_{si}, z_{si})$ is the position coordinate of the $i$th base station, $c$ is the propagation speed of electromagnetic waves, $t_{ik}$ is the time when the carrier signal reaches the $i$th base station at time $k$, $\delta t_{ik}$ is the delay error of the $i$th pseudolite at time $k$, $n_{ik}$ is the system noise at time $k$, $N$ is the number of visible base stations at time $k$ and $r_{ik}$ is the distance from carrier $k$ to the $i$th base station.

Suppose the measurement pseudoranges of $N$ base stations at time $k$ are $r_{1,k}, \ldots r_{i,k}, \ldots, r_{N,k}$, respectively. Then the position coordinate of the carrier at $k+1$ is $(x_{k+1}, y_{k+1}, z_{k+1})$ and the measured pseudoranges of $N$ base stations at time $k+1$ are $r_{1,k+1}, \cdots r_{i,k+1}, \cdots, r_{N,k+1}$, respectively.

Formula (1) is a nonlinear equation. The Taylor expansion of Formula (1) at position $(x_k, y_k, z_k)$ is as follows:

$$
\begin{pmatrix} r_{1k} - n_{1k} \\ \vdots \\ r_{ik} - n_{ik} \\ \vdots \\ r_{Nk} - n_{Nk} \end{pmatrix} = \begin{bmatrix} \frac{x_{s1}-x_k}{r_{1,k}} & \frac{y_{s1}-y_k}{r_{1,k}} & \frac{z_{s1}-z_k}{r_{1,k}} & 1 \\ \vdots & \vdots & \vdots & \vdots \\ \frac{x_{si}-x_k}{r_{i,k}} & \frac{y_{si}-y_k}{r_{i,k}} & \frac{x_{si}-y_k}{r_{i,k}} & 1 \\ \vdots & \vdots & \vdots & \vdots \\ \frac{x_{sN}-x_k}{r_{N,k}} & \frac{y_{sN}-y_k}{r_{N,k}} & \frac{z_{sN}-z_k}{r_{N,k}} & 1 \end{bmatrix} \begin{bmatrix} \Delta x_k \\ \Delta y_k \\ \Delta z_k \\ \Delta t \end{bmatrix}
\tag{2}
$$

where $\Delta x_k = x_{k+1} - x_k$, $\Delta y_k = y_{k+1} - y_k$, $\Delta z_k = z_{k+1} - z_k$.

$$
R = \begin{pmatrix} r_{1k} - n_{1k} \\ \vdots \\ r_{ik} - n_{ik} \\ \vdots \\ r_{Nk} - n_{Nk} \end{pmatrix}
\tag{3}
$$

$$
G = \begin{bmatrix} \frac{x_{s1}-x_k}{r_{1,k}} & \frac{y_{s1}-y_k}{r_{1,k}} & \frac{z_{s1}-z_k}{r_{1,k}} & 1 \\ \vdots & \vdots & \vdots & \vdots \\ \frac{x_{si}-x_k}{r_{i,k}} & \frac{y_{si}-y_k}{r_{i,k}} & \frac{x_{si}-y_k}{r_{i,k}} & 1 \\ \vdots & \vdots & \vdots & \vdots \\ \frac{x_{sN}-x_k}{r_{N,k}} & \frac{y_{sN}-y_k}{r_{N,k}} & \frac{z_{sN}-z_k}{r_{N,k}} & 1 \end{bmatrix}
\tag{4}
$$

$$
b = \begin{bmatrix} \Delta x_k \\ \Delta y_k \\ \Delta z_k \\ \Delta t \end{bmatrix}
\tag{5}
$$

Formula (2) can be written as:

$$
R = Gb
\tag{6}
$$

The data of each pseudosatellite base station of the least square algorithm can be solved. That is:

$$
b = \left( G^T G \right)^{-1} G^T R
\tag{7}
$$

The pseudosatellite receiver can receive the pseudorange and carrier phase from the pseudosatellite to the receiver. However, due to the complex indoor environment, serious multipaths and occlusion, the pseudorange and carrier noise ratio jump greatly in the indoor environment, so it is not suitable for indoor use. However, the carrier phase change rate of pseudolites arriving at the receiver is relatively stable, which can be used as the observation of indoor positioning.

In the $r_{ik+1}$ calculation, the pseudorange is not calculated directly but recursively through the carrier phase change, that is:

$$
r_{i(k+1)} = \lambda \left( N_{ik|i(k+1)} + \phi_{ik|i(k+1)} \right) + r_{ik}
\tag{8}
$$

where $\lambda$ is the wavelength of the carrier, $N_{ik|i(k+1)}$ is the whole cycle number of the carrier phase change in the $i$th pseudolite from $k$ to $k + 1$ and $\phi_{ik|i(k+1)}$ is the less-than-full-cycle part of the carrier phase change in the $i$th pseudolite from $k$ to $k + 1$.

In the indoor complex environment, the pseudorange can be directly used according to the actual environment or calculated through the carrier phase change rate so as to improve the navigation and positioning performance in the complex environment.

### 3.2. Wi-Fi/Bluetooth Positioning Principle

Wi-Fi/Bluetooth has the characteristics of low equipment cost and wide coverage. It is a positioning technology used more in indoor positioning. The geometric positioning method of Wi-Fi/Bluetooth mainly uses the RSSI attribute of Wi-Fi/Bluetooth and the propagation model of a Wi-Fi/Bluetooth signal in the air to carry out the inverse ranging of Wi-Fi transmission signal points [26] so as to realize trilateral positioning. However, the accuracy of energy positioning is low, and the positioning effect is poor. It needs a dense array. Combined with inertial sensors, it can realize navigation and positioning with certain accuracy, and the positioning effect is poor when used alone.

At present, the positioning method of multistation joint direction finding based on array Bluetooth is relatively widely used and has good navigation and positioning effect. The positioning principle based on Bluetooth arrays is to measure the azimuth of the carrier reaching each base station through multiple array base stations. In indoor complex environments, due to the limitation of array aperture and installation structure, the direction finding of Bluetooth arrays generally does not measure the pitch angle of the carrier and generally adopts the method of fixing the height of the carrier. The azimuth measurement adopts high-resolution direction-finding methods, such as MUSIC [27], ESPRIT [28] and other algorithms, which have good direction finding effects. After obtaining the direction finding results of multiple base stations, the least square method can be used to locate the carrier.

Then the least square positioning method based on the direction finding of multiple Bluetooth base stations can be described as:

$$s_{LS} = \left[ H^{\mathrm{T}} H \right]^{-1} H^{\mathrm{T}} X \tag{9}$$

where $H = \begin{bmatrix} \sin \varphi_1 & -\cos \varphi_1 \\ \vdots & \vdots \\ \sin \varphi_N & -\cos \varphi_N \end{bmatrix}$, $X = \begin{bmatrix} x_{B1} \sin \varphi_1 - y_{B1} \cos \varphi_1 \\ \vdots \\ x_{BN} \sin \varphi_N - y_{BN} \cos \varphi_N \end{bmatrix}$, $s_{LS} = \begin{bmatrix} x & y \end{bmatrix}^T$ is the carrier position coordinate, $\varphi_1 \cdots \varphi_N$ is the azimuth measurement result of $N$ base stations and $\begin{bmatrix} x_{B1} & \cdots & x_{BN} \\ y_{B1} & \cdots & y_{BN} \end{bmatrix}$ are the position coordinates of $N$ base stations.

### 3.3. Track Calculation Principle of Inertial Sensors

In recent years, due to the emergence of various miniaturized and low-cost inertial devices, the application field of track estimation based on inertial sensors has gradually expanded and can be applied to personnel navigation and positioning. In pedestrian navigation and positioning, the acceleration curve of pedestrian motion is similar to a sine wave, and each complete sine wave in the acceleration curve exactly corresponds to a walking cycle. Therefore, the statistics of steps are actually to identify and count the walking cycle from the acceleration curve and calculate the step size based on this. Therefore, correctly identifying the walking cycle is the key to the track calculation of inertial sensors.

$$\begin{cases} x_{k+1} = x_k + l_k \cos \alpha_k \sin \beta_k \\ y_{k+1} = y_k + l_k \cos \alpha_k \cos \beta_k \\ z_{k+1} = z_k + l_k \sin \alpha_k \end{cases} \tag{10}$$

where $(x_k, y_k, z_k)$ is the position coordinate at time $k$, $l_k$ is the step size of step $k$, $\alpha_k$ is the pitch angle at time $k$ and $\beta_k$ is the heading angle at time $k$. Pedestrian track estimation does not require high hardware requirements, and the calculation overhead is small, so it has good application value. However, due to the poor performance of the internal sensor devices of commercial mobile phones, it is easy to form a large deviation after a certain time accumulation and cannot be corrected. Therefore, other means are needed to correct the positioning information of PDR.

## 4. Elastic Fusion Location Algorithm

In the multiscene-switching and complex electromagnetic environment, the navigation and positioning signals received by the receiving terminal are not all direct signals. In the process of multisource fusion positioning, it is necessary to judge the availability of the current received signals and the participation rate of fusion positioning so as to complement each other and so as to achieve better navigation and positioning effects and realize seamless coverage indoors and outdoors. In the process of multisource fusion positioning, the selection of multisource data is very important. It is necessary to eliminate the positioning sources or positioning data greatly affected by the environment in order to achieve better navigation and positioning performance.

For homologous data, we can analyze whether there is a large jump in the data by comparing with the previous data so as to judge whether the current data are available, that is:

$$e_{mk} = \left\| \phi_{mk} - \phi_{m(k-1)} \right\|, \ e_{m(k-1)} = \left\| \phi_{mk(k-1)} - \phi_{m(k-2)} \right\| \tag{11}$$

$\phi_{mk}$ represents the observation of the $m$th sensor at time $k$ and $e_{mk}$ represents the observed change rate of the data of the $m$th sensor from time $k-1$ to time $k$.

Our hypothesis is:

$$Q_{mk} = \left\| e_{mk} - e_{m(k-1)} \right\| \tag{12}$$

Compare $Q_{mk}$ with the set threshold $\eta_T$. If $Q_{mk} \geq \eta_T$, it is considered that there is a jump in the data of the $m$th sensor at time $k$, which needs to be eliminated; otherwise, it is available. Threshold $\eta_T$ setting is related to carrier status and external environment.

In the multisource elastic fusion navigation algorithm, each navigation source is regarded as a variable node, and the variable node realizes local fusion at the function node. Finally, the navigation positioning of elastic fusion is realized by constructing a weighted constraint function.

Suppose the positioning result of the multisource sensor at time $k$ is $A_k = [a_{k1}, \ldots, a_{km}, \ldots a_{kM}]^T$, $M$ is the number of sensors and $a_{km} = \begin{bmatrix} x_{km} & y_{km} & z_{km} \end{bmatrix}$ is the positioning result of the $m$th sensor at time $k$.

Suppose the fusion location result at $k-1$ is $a_{k-1} = \begin{bmatrix} x_{k-1} & y_{k-1} & z_{k-1} \end{bmatrix}^T$. Set the threshold value $\eta_{th1}$ according to the carrier moving speed and positioning time. Calculate the distance between the positioning results of $M$ sensors at time $k$ and the fusion positioning results at time $k-1$, that is,

$$b_{km} = \sqrt{(x_{km} - x_{k-1})^2 + (y_{km} - y_{k-1})^2 + (z_{km} - z_{k-1})^2} \tag{13}$$

Compare $b_{km}$ with the set threshold $\eta_{th1}$. If $b_{km} - \eta_{th1} < 0$, it is considered that the positioning result of the $m$th sensor is available; if $b_{km} - \eta_{th1} \geq 0$, it is considered that the positioning result of the $m$th sensor is not available, so the location results of unavailable sensors in the fusion location are eliminated. In the fusion location, eliminate the location results of unavailable sensors, and change the number of available sensors to $M_1$.

Calculate the centroid $g_k$ of the positioning results of the remaining $M_1$ sensors, that is:

$$g_k = \begin{bmatrix} x'_k \\ y'_k \\ z'_k \end{bmatrix} = \begin{bmatrix} \frac{x_{k1} + \cdots x_{ki} + \cdots x_{kM_1}}{K} \\ \frac{y_{k1} + \cdots y_{ki} + \cdots y_{kM_1}}{K} \\ \frac{z_{k1} + \cdots z_{ki} + \cdots z_{kM_1}}{K} \end{bmatrix} \tag{14}$$

Calculate the distance $d_{ki}$ between the positioning results of the remaining $M_1$ sensors and the centroid $g_k$, that is:

$$d_{ki} = \sqrt{(x_{ki} - x'_k)^2 + (y_{ki} - y'_k)^2 + (z_{ki} - z'_k)^2} \tag{15}$$

Then define the weight of $k$-time fusion weighting as:

$$\omega_k^i = \frac{1}{\sqrt{2\pi}\sigma_i} e^{-\frac{d_{ki}}{2\sigma_i^2}} \tag{16}$$

where $\sigma_i$ is the mean square error of the positioning of the $i$th sensor.

Normalize the weight to obtain:

$$\overline{\omega}_k^i = \omega_k^i \left/ \sum_{i=1}^{M_1} \omega_k^i \right. \tag{17}$$

Weight the positioning results of the remaining $M_1$ sensors, that is:

$$a_k = \begin{bmatrix} x_k \\ y_k \\ z_k \end{bmatrix} = \begin{bmatrix} \overline{\omega}_k^1 x_1 + \cdots \overline{\omega}_k^i x_i + \cdots \overline{\omega}_k^{M_1} x_{M_1} \\ \overline{\omega}_k^1 y_1 + \cdots \overline{\omega}_k^i y_i + \cdots \overline{\omega}_k^{M_1} y_{M_1} \\ \overline{\omega}_k^1 z_1 + \cdots \overline{\omega}_k^i z_i + \cdots \overline{\omega}_k^{M_1} z_{M_1} \end{bmatrix} \tag{18}$$

According to Equation (18), the elastic fusion positioning result at time $k$ can be obtained. The implementation steps of the algorithm can be described as follows:

(1) Compare the data of the same sensor at the current time and the previous time to judge whether there is a large jump in the current data. If the jump is greater than the set threshold, it is considered that the current data are not available and eliminated during positioning; on the contrary, it is considered that the current data are available;

(2) Calculate the distance between the $k$-time positioning result of available multisource sensors and the $k - 1$-time fusion positioning result and compare it with the set threshold $\eta_{th1}$ to eliminate the positioning result greater than the threshold;

(3) Calculate the centroid $g_k$ of the remaining positioning results, and calculate the distance between the positioning results $d_{ki}$ of the remaining $M_1$ sensors and the centroid $g_k$;

(4) Construct the fusion weighting coefficient at the current time according to $d_{ki}$ and location variance $\sigma_i$, and normalize the fusion weighting coefficient;

(5) Weight the remaining $M_1$ sensor positioning results and fuse the results to obtain the fusion positioning results at the current time.

This paper proposes an algorithm based on the elastic fusion of multisource data to realize stable and continuous navigation and positioning in complex environments, which can solve navigation and positioning problem in indoor and outdoor complex environments, improve the stability and positioning performance of navigation systems in complex environments and is suitable for multisource fusion navigation and positioning in complex environments.

## 5. Results

### 5.1. Test Environment

The test site is the artificial intelligence test site of the State Key Laboratory of Satellite Navigation System and Equipment of the 54th Research Institute of China, Electronics Technology Corporation. Aiming at the problems of complex and changeable indoor environments, seamless and high-precision positioning and positioning based on radio signal under indoor multipath or occlusion conditions, a multisensor elastic fusion positioning model is constructed according to the physical characteristics of multisensors to realize submeter navigation and positioning. The test environment is shown in Figure 3:

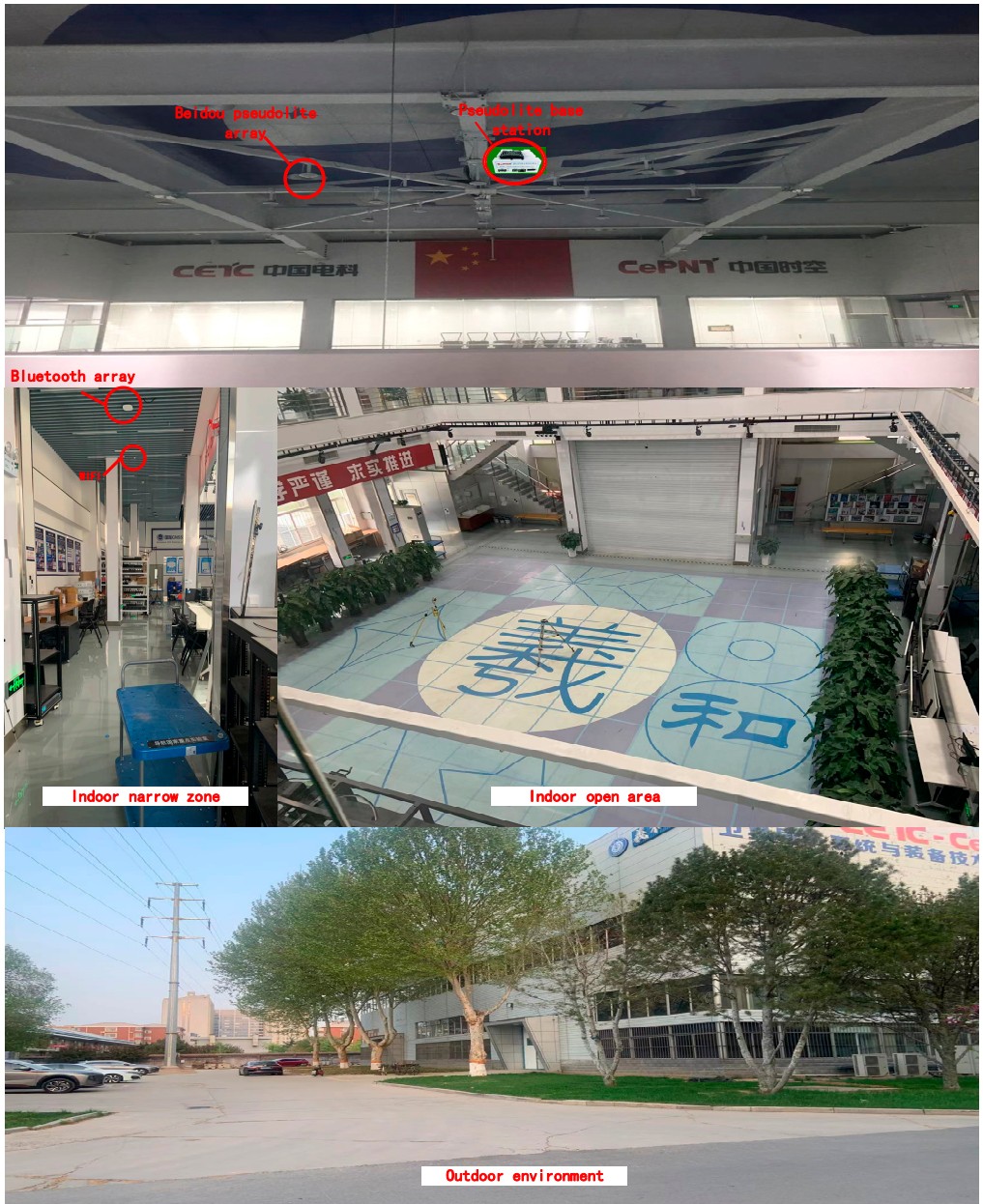

**Figure 3.** Test environment.

Figure 3 shows the indoor and outdoor environment of the artificial intelligence test field. The experimental environment includes an outdoor environment, an indoor open environment and an indoor narrow and long area. It involves the switching-of-positioning environment, which is more suitable for evaluating the algorithm proposed in this paper.

### 5.2. Indoor and Outdoor Static Positioning Tests

The northeast sky coordinate system is used to verify the static positioning accuracy of the algorithm in different regions. Five points are selected indoors and outdoors to test the navigation and positioning accuracy of the algorithm under static conditions, among which point 1 is outdoors, points 2 and 3 are in indoor open areas and points 4 and 5 are in indoor narrow areas. The test results are shown in Table 1.

**Table 1.** Static test.

| Serial Number | Coordinate Point | | Test Result 1's Deviations/m | Test Result 2's Deviations/m | Test Result 3's Deviations/m | Test Result 4's Deviations | Test Result 5's Deviations | Mean Square Error/m |
|---|---|---|---|---|---|---|---|---|
| 1 | X | 4,212,836.16 | 0.17 | 0.08 | 0.37 | −0.15 | −0.25 | 0.43 |
|   | Y | 538,261.31 | 0.37 | −0.14 | −0.14 | −0.56 | 0.27 | |
| 2 | X | 4,212,813.36 | 0.36 | 0.19 | −0.17 | 0.09 | −0.24 | 0.47 |
|   | Y | 538,258.63 | −0.49 | −0.42 | −0.30 | 0.35 | 0.5 | |
| 3 | X | 4,212,817.65 | 0.31 | −0.5 | −0.11 | −0.32 | 0.07 | 0.35 |
|   | Y | 538,266.43 | −0.31 | 0.08 | 0.13 | 0.16 | −0.08 | |
| 4 | X | 4,212,807.54 | −0.43 | 0.8 | 0.45 | −0.81 | −0.01 | 0.62 |
|   | Y | 538,276.63 | −0.32 | −0.2 | 0.22 | −0.11 | 0.08 | |
| 5 | X | 4,212,815.32 | 0.32 | 0.43 | 0.24 | 0.11 | −0.02 | 0.69 |
|   | Y | 538,278.52 | −0.40 | 0.92 | −0.89 | 0.45 | −0.20 | |

It can be seen from Table 1 that the static positioning accuracy is better than 0.5 m in outdoor areas and indoor open areas and better than 0.7 m in indoor narrow and long areas.

### 5.3. Dynamic Positioning Performance Test in Different Areas

(1) Pseudolite positioning test used in indoor open areas.

Figure 4 shows the navigation and positioning accuracy test of the circular pseudolite array. The red dot in the figure is the pseudolite currently selected for positioning, the blue line is the positioning track, and D1–D21 represent the location points where the pseudolite can be installed.

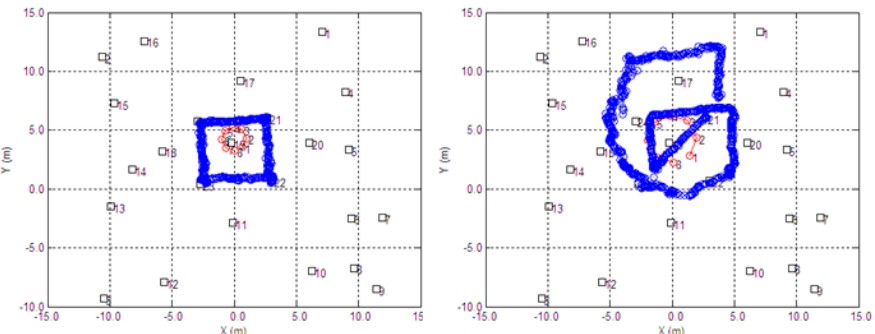

**Figure 4.** Pseudolite positioning in indoor open area.

It can be seen from Figure 4 that in the indoor open area, the pseudolite array has good navigation and positioning accuracy, the positioning trajectory and motion trajectory almost coincide and the average positioning accuracy calculated by postprocessing is better than 0.5 m.

(2) Positioning performance of Wi-Fi/Bluetooth and inertial sensor combination in indoor narrow and long areas.

Figure 5 shows the positioning performance of Wi-Fi/Bluetooth and inertial sensor combination in indoor narrow and long areas. The real track is a straight line, as shown by the red line in Figure 5.

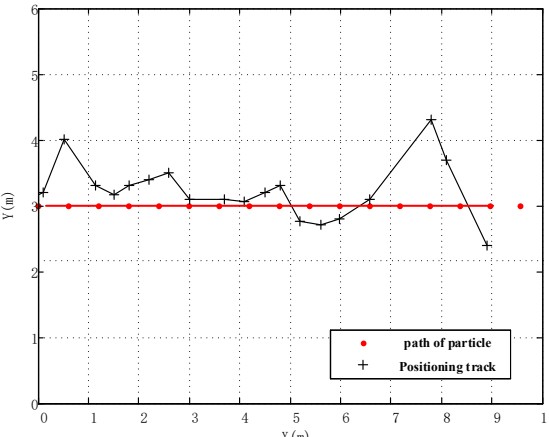

**Figure 5.** Combined positioning of Wi-Fi/Bluetooth and inertial sensor in indoor strip.

As can be seen from Figure 5, Wi-Fi/Bluetooth and inertial sensor combination positioning are adopted in the indoor strip, as shown by the black + line in Figure 5. The positioning accuracy is worse than that in the indoor open area, and the average positioning accuracy calculated by postprocessing is better than 0.7 m.

(3) Navigation and positioning performance of outdoor satellite navigation differential enhancement and 5G combination.

Figure 6 is the navigation and positioning performance test of the combination of outdoor environment differential enhancement and 5G in Figure 3.

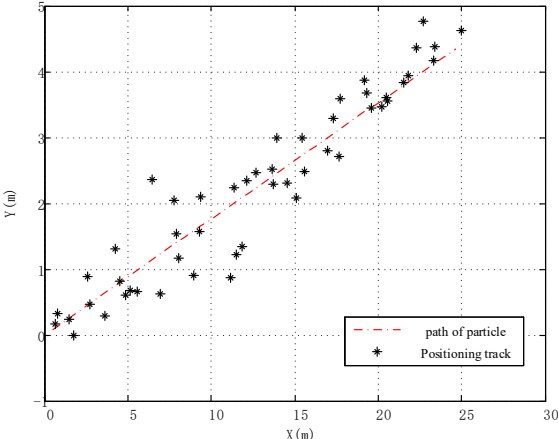

**Figure 6.** Navigation and positioning of outdoor satellite navigation differential enhancement and 5G combination.

It can be seen from Figure 6 that the positioning accuracy is better than 0.5 m by using the navigation and positioning mode of satellite navigation differential enhancement and 5G combination under the condition of unobstructed outdoors.

### 5.4. Indoor and Outdoor Continuous Elastic Navigation and Positioning Test

The elastic PNT positioning terminal developed by the State Key Laboratory of Satellite Navigation System and Equipment (as shown in Figure 2) is used to test the navigation and positioning performance in indoor and outdoor complex environments.

Firstly, the navigation and positioning performance of the proposed algorithm under fine motion is tested in the indoor open area. The test results are shown in Figure 7.

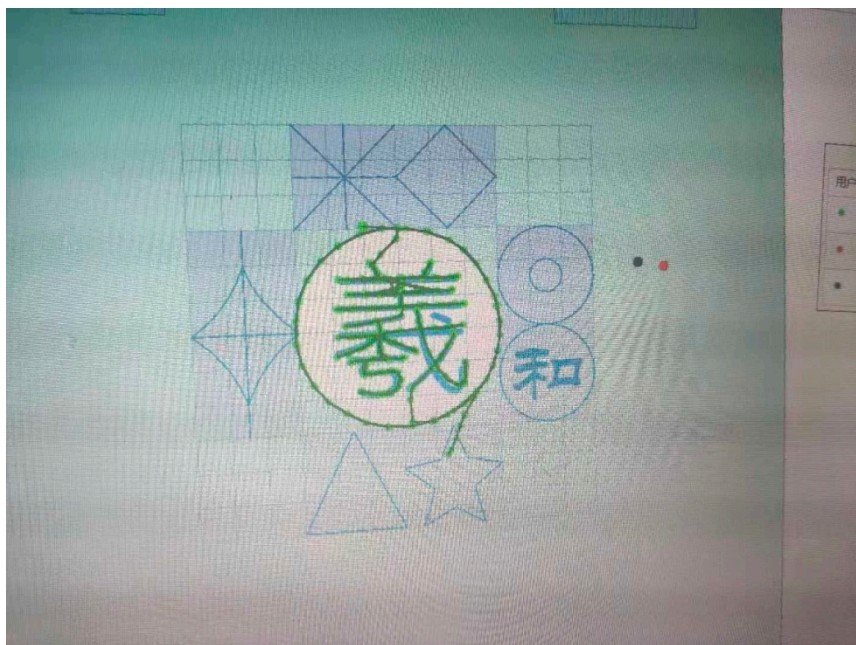

**Figure 7.** Dynamic test of indoor open area.

Figure 7 shows that in the indoor open area, the motion trajectory of the algorithm proposed in this paper forms a complex Chinese character. From the comparison between the motion trajectory and the background pattern in Figure 7, it can be seen that the motion trajectory and the background font pattern almost coincide, indicating that the algorithm proposed in this paper has high navigation and positioning accuracy in the indoor open area and can distinguish the subtle differences of the motion trajectory.

The performance of the algorithm proposed in this paper can be better verified under the dynamic condition of multiscene switching. Therefore, the multiscene switching dynamic positioning experiment is carried out in the indoor and outdoor environments of the artificial intelligence test field of the State Key Laboratory of Satellite Navigation System and Equipment. The test environment includes the indoor and outdoor environments, indoor narrow areas and indoor open areas. The test results are shown in Figures 8–10.

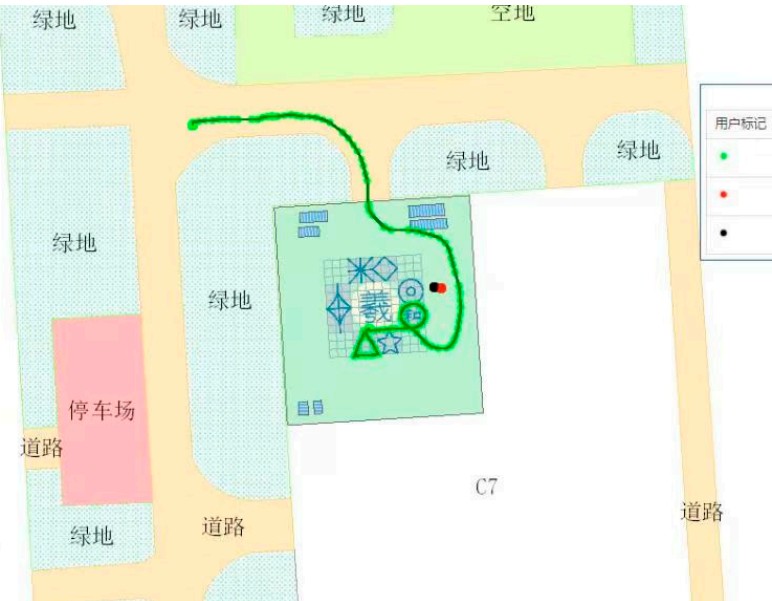

**Figure 8.** First test.

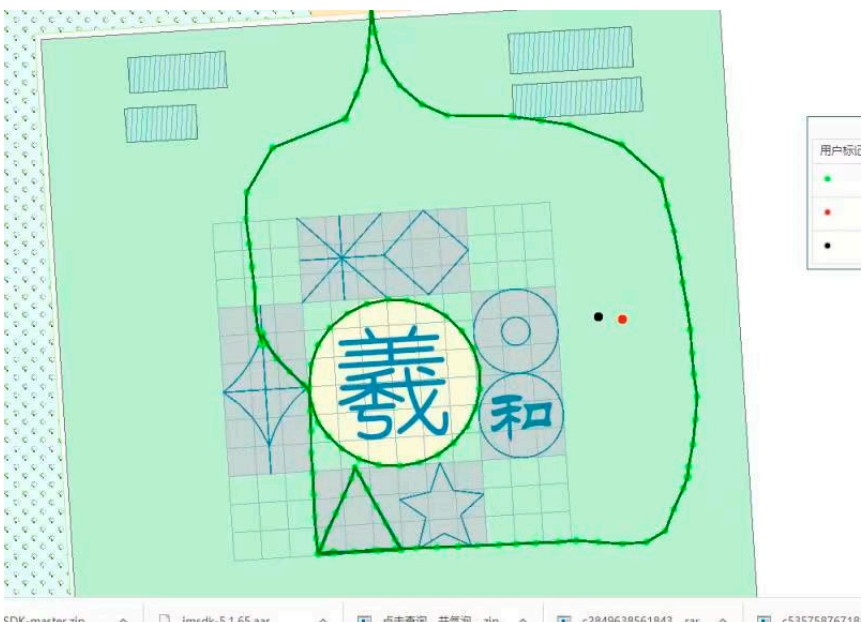

**Figure 9.** Second test.

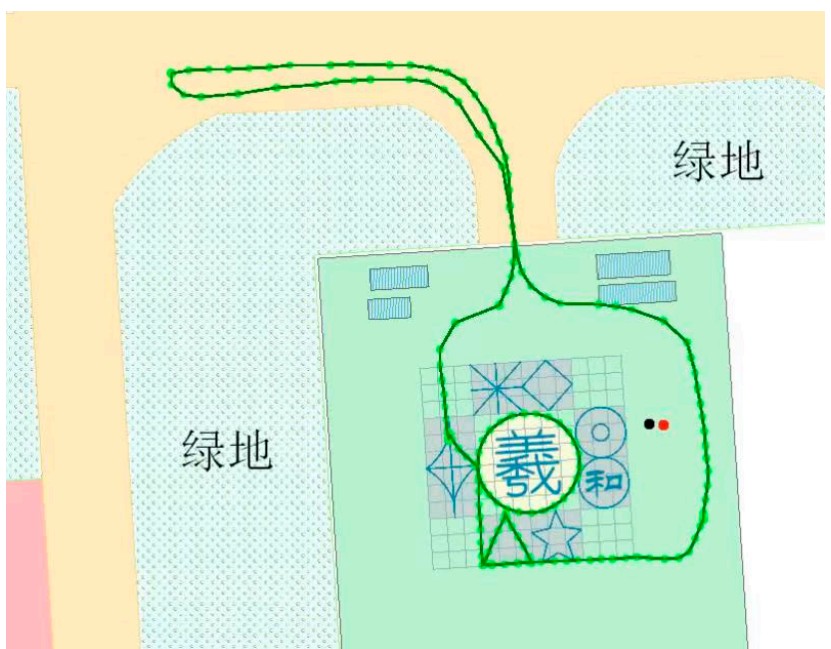

**Figure 10.** Third test.

In this test, the tester took the multisource elastic fusion positioning terminal from outdoors to indoors, passing through the outdoor area, indoor narrow areas and open areas. It can be seen from Figure 8 that the final trajectory can be in accordance with the figure on the ground.

Figure 9 shows that after the tester goes from the outdoors to the indoor narrow area, the indoor open area and then to the indoor narrow area, goes out of the room and returns to the starting point, the whole motion track is completely closed.

Figure 10 shows test personnel go from indoors to outdoors, starting from the indoor open area, passing through the indoor open area, the indoor long and narrow area, the outdoors, the indoor long and narrow area and the indoor open area to realize the complete closure of motion estimation.

From the dynamic test results in Figures 8–10, it can be seen that the navigation and positioning performance of the algorithm proposed in this paper is independent of the starting point position and can realize continuous and seamless navigation and positioning in complex and changeable environments.

Table 2 shows the average positioning accuracy of three dynamic tests in Figures 8–10.

**Table 2.** Average positioning error of dynamic test.

| | Path of Particle | | Average Positioning Error/m | |
|---|---|---|---|---|
| 1 | shown in Figure 8 | X | 0.4161 | 0.6372 |
| | | Y | 0.4826 | |
| 2 | shown in Figure 9 | X | 0.4322 | 0.6697 |
| | | Y | 0.5116 | |
| 3 | shown in Figure 10 | X | 0.4704 | 0.6846 |
| | | Y | 0.4974 | |

As can be seen from Table 2, the positioning accuracy of the three dynamic tests is better than 0.7 m. Compared with the static environment, the positioning accuracy is poor because the whole trajectory involves a variety of indoor and outdoor complex scenes and involves the adjustment and switching process, which affects the overall positioning performance. The navigation and positioning performance of the algorithm proposed in this paper can meet the current needs of high-precision indoor and outdoor navigation and positioning and has good stability. The positioning is relatively smooth in the scene-switching process without large jumps.

## 6. Conclusions

In this paper, we propose an indoor and outdoor multisource fusion navigation and positioning algorithm based on particle filters. According to the navigation and positioning requirements of indoor and outdoor complex scenes, this method uses a variety of sensor information to construct a multisource elastic fusion positioning model, eliminates abnormal positioning results according to the previous moment's positioning results and carrier motion state and calculates the centroid of the remaining multisource sensor positioning results. With consideration of the results from different sensors and the accuracy of different sensors, it shows that: the weights of different sensor results are given to achieve the flexible fusion localization of multisensor simultaneous interpreting.

The test environment is built in the artificial intelligence test field of the State Key Laboratory of Satellite Navigation System and Equipment, and the multisource elastic fusion positioning terminal developed by the State Key Laboratory of Satellite Navigation System and Equipment is used to test and verify the positioning performance of the method proposed in this paper. The test results show that the proposed algorithm can solve the problem of seamless navigation and positioning under the condition of indoor and outdoor multiscene switching. The elastic fusion positioning accuracy is better than 0.7 m, and the positioning stability is good. There are no jumps in the positioning results in the three tests.

This paper proposes an algorithm to solve the problem of indoor and outdoor high-precision seamless navigation and positioning continuity. The redundant information of elastic multisource sensors is used to improve the robustness of navigation and positioning in complex and changeable scenes. In the future, it can be applied to positioning and navigation, personnel management, safety rescue and other aspects in complex urban scenes, promote the all-round development of the indoor positioning blue ocean market and finally realize the seamless integration of the indoor and outdoor interconnection of all things.

**Author Contributions:** All authors contributed to the manuscript and discussed the results. All authors together developed the idea that led to this paper. G.F. and C.S. conceived the experiments and analyzed the data. B.Y. provided critical comments and contributed to the final revision of the paper. L.H. contributed to the expression and the design of programs. Q.R. wrote the manuscript and all the authors participated in amending the manuscript. All authors have read and agreed to the published version of the manuscript.

**Funding:** This work was supported in part by the National Key Research and Development Plan of China (project: Intelligent Emergency Rescue Equipment Platform Technology and Equipment: No. 2019YFC1511500 and Indoor Hybrid Intelligent Positioning and Indoor GIS Technology: No. 2016YFB0502100).

**Data Availability Statement:** Not applicable, the study does not report any data.

**Conflicts of Interest:** The authors declare no conflict of interest.

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
