# Peer review of "An Indoor and Outdoor Multi-Source Elastic Fusion Navigation and Positioning Algorithm Based on Particle Filters"

_futureinternet, doi:10.3390/fi14060169_

Round 1

Reviewer 1 Report

  • It would be useful to describe how the thresholds n_T and n_th1 are derived or calculated
  • It is mentioned that if the sensor's observation difference is greater than the threshold n_T, it would be eliminated.  The question is then if the current observation is eliminated, then what about the next observation?  The device would have moved further at the next time step, thus the observation difference would be greater and would likely to be greater than the threshold n_T.  How would you handle this?
  • In the testing, good to include description on how did you get the ground-truth
  • In Table 1, what is the coordinate system used?  It does not look like longitude and latitude
  • In figure 4, good to label what are the red and blue points/lines.  Which one is the ground-truth?

Author Response

Point 1:  It would be useful to describe how the thresholds n_T and n_th1 are derived or calculated.

Response 1: Threshold n_ T and n_ Th1 is set according to the motion state and operating environment of the carrier. This value is an empirical value related to the motion state and operating environment of the carrier.

Point 2: It is mentioned that if the sensor's observation difference is greater than the threshold n_T, it would be eliminated.  The question is then if the current observation is eliminated, then what about the next observation?  The device would have moved further at the next time step, thus the observation difference would be greater and would likely to be greater than the threshold n_T.  How would you handle this?

Response 2: The threshold is set according to the motion state of the carrier and the position of the previous time, which has nothing to do with how far the carrier moves, but the premise is that the positioning result of the previous time is assumed to be within a reasonable error range.

Point 3: In Table 1, what is the coordinate system used?  It does not look like longitude and latitude.

Response 3: The northeast sky coordinate system is used in Table 1

Point 4: In figure 4, good to label what are the red and blue points/lines.  Which one is the ground-truth?

Response 4: The red line adopts the pseudo satellite position, and the blue line is the motion track

Reviewer 2 Report

Dear authors,

I have read your paper. The research is interesting, but from my point of view, the theory is really short, more details can improve the paper.

Table 1 - better is to use a local navigation frame/system, such as North-East-Down. Also for all results, I would like to see deviations/errors, not only one Mean Square Error.

Fig. 4 - The points with numbers in the figure should be explained. What are these points?

Fig. 5 - The red line says, that value is zero. You have a lot of empty space in the figure. Also, you can use the right vertical axis for red results. The reader can see the line progress.

Fig.6 - Legend is missing. It should be added.

Table 2 - The resolution of numbers 10-4 does not make sense.

Fig. 7 - Fig. 10 - The dynamic tests are fine, but the calculated trajectory in all dimensions should be done, also the error figure, etc. More analyses should be added. The research is complex and more results, and more details about results and accuracy analyses should be done. Not only one number as a result of the dynamic test.

Best regards,

Reviewer

Author Response

Point 1:  Table 1 - better is to use a local navigation frame/system, such as North-East-Down. Also for all results, I would like to see deviations/errors, not only one Mean Square Error.

Response 1: Thank you very much for your review. I have revised and improved my formulation and marked some changes in yellow.

Point 2: Fig. 4 - The points with numbers in the figure should be explained. What are these points?

Response 2: I have explained the number points in the figure and marked them in yellow

Point 3: Fig. 5 - The red line says, that value is zero. You have a lot of empty space in the figure. Also, you can use the right vertical axis for red results. The reader can see the line progress.

Response 3: I have modified it

Point 4: Fig.6 - Legend is missing. It should be added.

Response 4: I have added the  Legend in figure 6.

Point 5: Fig. 7 - Fig. 10 - The dynamic tests are fine, but the calculated trajectory in all dimensions should be done, also the error figure, etc. More analyses should be added. The research is complex and more results, and more details about results and accuracy analyses should be done. Not only one number as a result of the dynamic test.

Response 5: The dynamic test in Figure 7-10 uses the software screenshot of the elastic navigation and positioning terminal. At present, the terminal software cannot generate real-time errors, and detailed analysis can be carried out in the next work.

Round 2

Reviewer 2 Report

Dear authors,

thank you for the revision, you have done.

Best regards,

Reviewer